# Genome-Wide Association Study for Body Conformation Traits and Fitness in Czech Holsteins

**DOI:** 10.3390/ani12243522

**Published:** 2022-12-13

**Authors:** Jindřich Čítek, Michaela Brzáková, Jiří Bauer, Ladislav Tichý, Zuzana Sztankóová, Luboš Vostrý, Yvette Steyn

**Affiliations:** 1Department of Genetics and Agricultural Biotechnology, Faculty of Agriculture, University of South Bohemia in České Budějovice, Studentská 1668, 370 05 České Budějovice, Czech Republic; citek@fzt.jcu.cz; 2Veterinary Research Institute, Hudcova 296, 621 00 Brno, Czech Republic; 3Institute of Animal Science, Přátelství 815, 104 00 Praha, Czech Republic; tichy.ladislav@vuzv.cz (L.T.); sztankoova.zuzana@vuzv.cz (Z.S.); 4Czech Moravian Breeders’ Corporation, Benešovská 123, 252 09 Hradištko, Czech Republic; bauer@plemdat.cz; 5Faculty of Agrobiology, Food and Natural Resources, Czech University of Life Sciences Prague, 165 00 Praha, Czech Republic; vostry@af.czu.cz; 6Department of Animal and Dairy Science, University of Georgia, 425 River Road, Athens, GA 30602, USA; yvette.steyn@uga.edu

**Keywords:** cattle, Holstein, body conformation, GWAS, wssGBLUP, SNP, dairy capacity, feet and legs

## Abstract

**Simple Summary:**

The ideal body conformation is an important part of the breeding objective of Czech Holsteins to allow high production while improving health and fitness. In this study, we aimed to identify loci that influence these traits. The genome-wide association study (GWAS) was performed using the weighted single-step best linear unbiased prediction (wssGBLUP) method. The multiple comparison test was performed with the Bonferroni correction. Composite traits (dairy capacity composite, feet and legs composite, and total score) and partial linear traits (stature, body depth, angularity, and fore udder attachment) each showed associations with one single nucleotide polymorphism (SNP) that were either statistically significant or approached the significance threshold. The association analysis without the Bonferroni correction (based on the significance level of 1.00 × 10^−6^) showed one significant SNP for total score, and also one for stature, angularity, and fore udder attachment. Moreover, one SNP was of near-significance threshold for the dairy capacity composite, the feet and legs composite, and body depth.

**Abstract:**

The aim of this study was a genome-wide association study (GWAS) on conformation traits using 25,486 genotyped Czech Holsteins, with 35,227 common SNPs for each genotype. Linear trait records were collected between 1995 and 2020. The Interbull information from Multiple Across Country Evaluation (MACE) was included for bulls that mostly had daughter records in a foreign country. When using the Bonferroni correction, the number of SNPs that were either significant or approached the significance threshold was low—dairy capacity composite on BTA4, feet and legs composite BTA21, total score BTA10, stature BTA24, body depth BTA6, angularity BTA20, fore udder attachment BTA10. Without the Bonferroni correction, the total number of significant or near of significance SNPs was 32. The SNPs were localized on BTA1,2,4,5,6,7,8,18,22,25,26,28 for dairy capacity composite, BTA15,21 for feet and legs composite, BTA10 for total score, BTA24 stature, BTA6,23 body depth, BTA20 angularity, BTA2 rump angle, BTA9,10 rear legs rear view, BTA2,19 rear legs side view, BTA10 fore udder attachment, BTA2 udder depth, BTA10 rear udder height, BTA12 central alignment, BTA24 rear teat placement, BTA8,29 rear udder width. The results provide biological information for the improvement of body conformation and fitness in the Holstein population.

## 1. Introduction

Complex traits are an enigma [1] that concern both animals and humans. Even though numerous quantitative trait loci (QTLs) underlying complex traits were described, the respective mutations remain unknown. Fortunately, the latest results in genome sequencing, namely the localization of many single nucleotide polymorphisms (SNPs) and their genotyping in microarrays, have enabled the possibility of a new progress in gene discovery. This approach, especially when used in genome-wide association studies (GWASs), promises new perspectives in the describing of plenty of genes controlling complex traits in the near future. It is crucial for studies of host resistance to infectious diseases and important in both animal and human health [1]. Great progress has been made in GWASs in animal breeding. Genes important for economically relevant traits have been identified. Association studies in livestock should work toward the identification of causative mutations for traits of great economic importance. Such research will ultimately contribute to the understanding of the genetic control of polygenic traits in livestock. In such a way, the improvement of animal breeding will be feasible [2].

Twenty years ago, the fundamentals of genomic selection (GS) were described [3]. It was hypothesized that the application of GS would lead to a significant rise in the tempo of the breeding progress. Genomic selection gave rise to a paradigm change in breeding practice. The dairy industry in particular met the necessary conditions to achieve results [4,5]. The application of genomic selection has become a popular tool in the dairy industry and led to substantial increases in genetic gain [6]. Command of important genes and haplotypes, including their regulatory mechanisms as markers for quantitative traits, may improve strategies for dairy cattle selection in the present and future [7]. Here, GWAS plays a key role. The GWAS analyses of diverse dairy traits were conducted to identify important QTLs and SNP markers [8]. To date, the animal QTLdb has systematized 983 publications and 130,407 QTLs for cattle, the largest in livestock species. Most SNPs with the described associations with traits appear to be in linkage disequilibrium with a hitherto unknown causative mutation. The identification of functionally relevant DNA mutations is necessary for efficient genomic selection. This concept presumes that a high number of genes must be known for each known QTL. All types of DNA polymorphisms and epimutations must be examined to achieve outstanding genetic progress during selection [9]. Analyses of SNPs allow the detection selection signatures, such as specific shift in the frequency alleles and haplotypes frequency, as well as an increase or reduction in genetic diversity. This helps to identify modifications in the cattle genome in response to natural and artificial selection, and also loci and genetic variants directly affecting traits important for adaptation and production [10,11,12,13,14].

Stature is affected by many polymorphous genes of small effects in human beings. On the contrary, variation in dogs, even within breeds, has been suggested to be largely due to variants of a low number of genes [15]. The authors used data from cattle comparing the genetic architecture of stature to those in human beings and dogs and performed a meta-analysis for stature analyzing 58,265 cattle from 17 populations with 25.4 million imputed whole-genome sequence variants. Their results demonstrated that the genetic control of stature in cattle is similar to that in human beings, as the main variants in 163 significantly associated regions of genome explained at most 13.8% of the phenotypic variance. These variants were mostly non-coding. There was considerable overlap in loci for stature with human beings and dogs, proposing that in mammals a set of common genes regulates body size. Other authors’ results also suggest that, despite QTLs specific for breed and species, the genetic structure of body mass may have been conserved in mammals by the same evolutionary forces [16]. However, the genetic heterogeneity that occurs when diverse genetic compositions underlying various populations result in the same phenotypes must also be considered [17]. Analyzing the genetic control of body size is critical for cattle breeding to improve both efficiency and productivity [18].

In a multitrait meta-analyses performed in Brown Swiss cattle, the most significant SNPs for body size were found on BTA1, 3, 11, and 26; on BTA3, 13, and 26 for leg conformation; on BTA3, 5, 6, 17, and 19 for mammary gland morphology; on BTA5 and 25 for body conformation; and on BTA10 for growth and carcass quality [19]. They located the lead SNP for body size in the third intron of the *BTRC* gene, which codes a member of the F-box protein family. The gene is involved in Wnt signaling, which plays a critical role in development and has been shown to be associated with process of limb development [20]. Another study also found a strong pleiotropic locus affecting milk yield, fat and protein yield, the lactating cow’s ability to cycle after calving, and the stature and body depth in Brown Swiss cattle on BTA25. Furthermore, very interesting signals for angularity were found on BTA11. Many of these signals overlapped with previously described QTLs for related traits in dairy and beef cattle [21]. SNPs associated with growth traits were found in Braunvieh on BTA22, 11, and 27 [22]. An et al. used the Illumina Bovine HD 770 K BeadChip to apply a GWAS to evaluate body size as abdominal size, heart size, hip height, body height, body length, and cannon bone size. They identified suggestive candidate genes on BTA1, 2, 3, 6, 7, 8, 9, 10, 11, 13, 19, and 23. Among these, 21 genes were full of promise candidate genes [23]. One of the regions of interest was on BTA14 and contains the *XKR4*, *TMEM68*, *TGS1*, *LYN*, *RPS20*, *MOS*, *PLAG1*, and *CHCHD7* genes, which are well-known candidates for feed intake, carcass-related traits, and growth [24]. The *PLAG1* gene (or *PLAG1-CHCHD7* region) is a strong candidate liable for body size, including stature, height, and body mass in many cattle breeds [25]. The polymorphic SNP BovineHD1400007259, located in the intron region of the *PLAG1* gene, is regarded a causal mutation responsible for stature [26].

A GWAS conducted in Chinese Holsteins revealed 11 SNPs associated with body-shape traits on 9 chromosomes, namely, BTA3, BTA4, BTA6, BTA7, BTA12, BTA13, BTA20, BTA22, and BTA29 [27]. A medium heritability of 0.20 to 0.38 was found. Other authors found 59 genome-wide significant SNPs associated with 26 conformation traits of 29 evaluated in the same breed; 5 SNPs were within earlier reported QTL regions, and 11 were in close proximity to the reported SNPs [28]. Twenty-two SNPs were located within known gene regions, while the other were 0.6–826 kb away from known genes. Additionally, four SNPs were found that influenced four pairs of traits, and the genetic correlation between each pair of traits ranged from 0.35 to 0.86, indicating that these SNPs may have a pleiotropic effect. Zhang et al. described 27 SNPs significantly associated with hip height and heart girth at different growth stages of Holsteins and 66 candidate genes located near the associated SNPs. Of these SNPs, nine genes are known to be highly related to development, including skeletal and muscular growth [18]. Compared with the single trait system, a multi-trait analysis increased the power to detect associations between SNPs and body composition traits in sheep [29]. A group of 23 SNPs affected mature size based on their pattern of effects across traits. However, the genes near this group of SNPs did not share any obvious function.

The objective of this study was to identify loci determining conformation traits utilizing a genome-wide association study in Holstein cattle in the Czech Republic. There are two main dairy breeds in the country, namely Czech Spotted, which is a part of Simmental group, and Holstein. Because the GWAS was not conducted in the Czech dairy cattle, this paper should fill up the gap.

## 2. Materials and Methods

Animal Care and Use Committee approval was not required for this research because the data were obtained from an existing database supplied by the Czech-Moravian Breeder´s Corporation (Czech Republic) and the Holstein Cattle Breeders Association of the Czech Republic.

### 2.1. Phenotypic Data

The analyzed dataset contains linear type trait records from 699,681 primiparous Holstein cows collected between 1995 and 2020. Linear-type trait evaluation was assessed according to the World Holstein Friesian Association on a 9-point scale by qualified classifiers. Cows are evaluated only once in the period 30 to 210 days after first calving, with their age at first calving between 600 and 1004 days.

The body score was evaluated as a part of a national cattle breeding program in the Czech Republic. There are 20 linear type traits, i.e., stature, chest width, body depth, angularity, rump angle, rump width, rear legs rear view, rear legs side view, foot angle, fore udder attachment, front teat placement, teat length, udder depth, rear udder height, central ligament, rear teat placement, rear udder width, bone quality, locomotion, and body condition score. These traits were also grouped into five composite traits. These composites are body conformation (rump angle, rump width, stature, body depth, chest width), foot and leg conformation (rear legs—rear and side view, foot angle, bone quality, locomotion), udder conformation (udder depth, rear udder height, front and rear teat placement, central ligament, fore udder attachment, teat length, and rear udder width), dairy capacity (angularity, bone quality, chest width, body depth, and stature), and total score. The total score is a weighted sum of the composite traits compiled as follows: 25% body conformation, 15% foot and leg conformation, 20% udder conformation, and 40% dairy capacity).

The Czech Holstein population is strongly connected to the populations from other countries. To increase the amount of information for breeding bulls used in the domestic population, Interbull information from Multiple Across Country Evaluation (MACE) was added according to the methodology from Přibyl et al. [30]. The effective record contribution of Interbull bulls were included if most of the daughters of bulls related to the domestic population have performance records in a foreign country that have a strong correlation with our population (USA, CAN, DEU, FRA, ITA, ESP, BEL, NLD, DNK, FIN, SWE) and the bull´s GEBV reliability was higher than 0.5.

### 2.2. Genotyping

The Czech Holstein population was genotyped throughout the whole country. From the population, all bulls used in reproduction were genotyped. In the past, the number of genotyped females was low. The milestone was the year 2018 when extensive genotyping of heifers began. For this reason, 98% of included genotypes are from young animals and these animals have a very high genetic connectedness in the population.

A total of 5419 bulls and 20,067 cows included in this study were commercially genotyped by various SNP chips (Table 1). All available SNP chips were included without any preselection, but only SNP chips with at least 60% of the common SNPs with Illumina 50 k v2 were included for further analysis. Quality control included the removal of markers on sex chromosomes, SNPs not in Hardy–Weinberg equilibrium, a minor allele frequency lower than 0.05, a call rate below 90%, and parent–progeny conflicts. The remaining data included 25,486 genotyped individuals with 35,227 common SNP markers.

### 2.3. GEBVs Prediction

In our analysis, the wssGBLUP method was used to predict genomic breeding values for all 25 linear-type traits using the BLUPF90 software suite [31]. Our study followed the studies of Němcová et al. [32], where the genetic parameters and heritability coefficients were estimated, and Zavadilová et al. [33], where the single-step genomic evaluation for linear type traits was developed. In our study, linear type traits were evaluated separately (single-trait), and correlations between traits were not considered because of the large size and computational demand of a multi-trait evaluation.

Genomic breeding values were predicted based on the model:**y** = **Xβ** + ***Zu*** + ***e***
where **y** is the vector of performances (a linear-type trait of females); **β** is a vector of fixed effects including the contemporary group (herd-year-month of classification and classifier) with at least two cows in the group, a classifier (fixed class effect from 1 to 4), linear and quadratic regression for cow´s age, and linear and quadratic regression of DIM (days in milk at linear classification); ***u*** is a vector of random additive genetic effects (6 generation pedigree with a total of 1,441,276 individuals were included in the relationship matrix ***H***); **X** is the incidence matrix relating phenotypes to the fixed effects; ***Z*** is the design matrix relating phenotypes to breeding values (***u***); and ***e*** is a vector of random residual error.

The relationship matrix ***H*** was created as a combination of the relationship matrix ***A*** (based on pedigree for all individuals) and the relationship matrix ***G*** (only for genotyped individuals), which is also combined with the relationship matrix ***A*_22_** (based on pedigree for genotyped individuals only):
$$H^{-1}=A^{-1}+\left[\begin{array}{cc} 0 & 0\\ 0 & G^{-1}-A_{22}^{-1} \end{array}\right]\;$$


We assumed the normal distribution of the additive genetic effect 
$N\left(0,H\sigma_{u}^{2}\right)$
 and random residual error 
$N\left(0,I\sigma_{e}^{2}\right)$
 and a zero covariance between them.

### 2.4. Genome-Wide Association Study

Genomic control of the population structure was performed by the genomic inflation factor (λ). Quantile-quantile (Q-Q) plots and Manhattan plots were created using R software version 4.2.2. (snpStats and qqman package) [34]. The genomic inflation factor was estimated based on the SNP *p*-values. The genome-wide association study (GWAS) was performed using the wssGBLUP method. The weights were estimated for all 35,227 SNPs using postGSf90 software [35,36]. There are seven steps during the iteration process [36,37]:(1)Creation of an identity matrix of SNP weights (***D*** = ***I***) where *d_i_* is the *i*th diagonal element of ***D*** which represents the variance of the SNP effect:

$$\mathrm{var}(s)=D=\left[\begin{array}{cccc} d_{1} & 0 & \ldots & 0\\ 0 & d_{2} & \ldots & 0\\ \ldots & \ldots & \ldots & \ldots \\ 0 & 0 & \ldots & d_{m} \end{array}\right].$$

(2)Calculation of the ***G*** matrix [38]:

$$G=\frac{{ZDZ}^{\prime }}{2\sum p_{i}\left(1-p_{i}\right)}\;$$
where ***Z*** is a matrix of centered genotypes, ***D*** is a diagonal matrix of weights, and 
$p_{i}$
 is the allele frequency of SNP *i*
(3)GEBV calculation to obtain the direct additive effect of individuals (***û***)(4)Decomposition of the GEBV (***û***) into the SNP effect (***â***)where ***D*** is a diagonal matrix of weights, ***Z*** is a matrix of the centered genotypes, 
$\hat{u}$
**_g_** is a vector of GEBV (only genotyped individuals)
(5)SNP variance estimation for each SNP. The non-linear A method was used [38,39]:

$$d_{i}=1.125^{\frac{\left|{\hat{a}}_{i}\right|}{\sigma \left(\hat{a}\right)}-2}\;$$
where the value 1.125 is a constant that describes the deviation of the SNP effect from a normal distribution, 
$\left|{\hat{a}}_{i}\right|$
 is the absolute estimated SNP effect for marker *i*, and *σ*(***â***) is the standard deviation of the vector of the estimated SNP effects.

(6)Normalization of matrix ***D*** and construction of matrix ***D*** based on the estimated SNP weights.(7)Running of the next iteration or stopping the loop. Only two iterations were used in our procedure to maximize accuracy [40]. The estimated weights were used in the GWAS analysis.

The multiple comparison test was performed by the Bonferroni correction. The significance threshold was determined as 0.05 divided by the number of SNPs used for the significance threshold and 0.01 divided by the number of SNPs used for the highly significance threshold.

## 3. Results and Discussion

The basic statistics of the predicted genomic breeding values are shown in Table 2 for all 25 linear type traits. The heritability of traits ranges from 0.07 to 0.49 depending on the selected trait.

The genomic inflation factor (λ) was lower than 1.1 for the most of traits (Appendix A). However, a λ higher than 1.1 was also observed for stature, rear legs rear view, rear teat placement, rear udder width, body condition score, composite body weight score, composite conformation score, and composite dairy capacity score where it even reached a value 12.076. The results of the Q-Q plots reveal that there could be some population stratification affecting the GWAS analysis. The very high value of genomic inflation factor may be caused also by factors such as high linkage disequilibrium, the strong association between phenotypic traits, and SNPs or systematic technical bias [41]. Our GWAS analysis included 25,486 genotypes collected during a long period. However, 98% of genotypes originated from young animals. Our population is open and therefore could be affected by the different bull´s subpopulations from foreign countries. However, the influence of these bulls accompanied the entire development of the breed, and we assumed that the differences in allele frequencies were not high due to the low genetic diversity of Holstein breed worldwide.

The significant SNPs were visualized using Manhattan plots. The level of significance was expressed by –log10 of the SNP´s *p*-value. The statistical significance was performed on the two levels, 1.00 × 10^−6^ for the suggestive significance threshold and 1.00 × 10^−8^ level for the genome-wide significance threshold. Manhattan plots are shown in the Appendix A.

A total of twenty linear traits and five linear composite traits were evaluated in the GWAS. After the Bonferroni correction, only four SNPs had statistically significant associations with the linear type traits, namely, fore udder attachment (highly significant, BTA10), stature (BTA24), angularity (BTA20), and the total score (BTA10) (Table 3, Appendix A). Body depth, dairy capacity composite, and feet and leg composite each had one SNP that approached the significance threshold (significance criteria are provided in Table 3).

Multiple testing was performed by the Bonferroni correction with the critical values as α_EW_ = α/k, where k was 35,227, and α was 0.05 or 0.01, representing the significance level (*p ≤* 1.42 × 10^−6^) and high significance level (*p ≤* 2.84 × 10^−7^), respectively. The computation of the Bonferroni correction is simple, but the correction is conservative, and the portion of the false null hypotheses that are correctly rejected is significantly reduced [42]. With the increasing number of hypothesis tests (k), the critical value rapidly decreases. For example, considering the 100 SNPs in the study, the thresholds determined by the Bonferroni correction could be 18 times lower than those determined by the Benjamini and Yekutieli method (modified FDR) [42]. Therefore, some authors recommended other multiple testing procedures, e.g., False Discovery Rate (FDR) [43]. Despite this recommendation, the Bonferroni correction is still used in the GWAS analyses. For this reason, we also presented SNPs that are at the suggestive 1.00 × 10^−5^ level.

The single-step GBLUP approaches are widely used for GEBV predictions. The advantage of the single-step methodology is the ability to include genotyped and non-genotyped individuals together in GEBV prediction, which allows for the inclusion of more information to provide better GEBV prediction accuracy [36,44]. Biologically, genetic variance is not equal across markers, and some major genes exist in the genome [38]. The genetic variance caused by individual SNPs could be estimated by weighted analysis, where weights (explained variance by each SNP) are determined for each SNP. The GEBV prediction incorporating these weights resulted in better accuracy because of including the linkage equilibrium between the SNP and causal genes, genetic links through unknown common ancestors, and Mendelian segregation [45].

When only a few QTLs are included, the GEBV prediction accuracy increases with the increasing number of genotyped individuals, but the differences between weighted and unweighted analysis become negligible when the number of QTLs and genotyped individuals is high [46]. Nevertheless, the advantage of the weighted analysis is the possibility of using the weights for the GWAS analysis [28,47,48]. Furthermore, with the increasing number of QTLs and genotyped individuals, the identification of QTLs is more accurate, which allows for clearer Manhattan plots due to better resolution [46]. Nevertheless, the localization ability decreases at the same time [49]. The benefits of weighted analysis were maximized using two iterations [40]. However, the power of the method could limit QTL identification when the effect is small [46].

Even though our number of genotyped individuals exceeded 25,000 and the number of SNP exceeded 35,000, only four significant SNPs were found for 25 linear-type traits. The low number of significant SNPs may be due to the polygenic character of linear-type traits, the strict *p*-value threshold using the Bonferroni correction or chosen SNPs did not cover most of the significant SNPs in the genome.

A somewhat less stringent method of testing without the Bonferroni correction did not show more significant associations with SNPs (Table 4, Appendix A). Again, total score, stature, angularity, and fore udder attachment had significant associations without the Bonferroni correction, namely with the same SNPs as with the Bonferroni correction.

The next 25 SNPs were observed on the statistical threshold of *p* < 1.00 × 10^−6^, and three SNPs were observed on the significance threshold of *p* < 1.00 × 10^−5^. Thirteen of these SNPs were associated with the dairy capacity composite; the SNPs were located on twelve chromosomes, and two SNPs with the feet and legs composite. Six SNPs were associated with udder conformation traits (udder depth, rear udder height, central alignment, rear teat placement, rear udder width), four with foot and leg conformation (rear legs rear and side view), and three with body conformation (body depth, rump angle). The dairy capacity composite is composed of linear-type traits that are related to an assumption of adequate feed intake and high milk production. Milk production is a polygenic trait, but the presence of major genes has been confirmed by many authors [50,51,52].

Our SNPs for the linear composite traits were found on different chromosomes compared to other studies. Wu et al. [28] reported the SNPs for dairy character on chromosomes 3, 12, 16, while Cole et al. [53] reported the SNP on chromosomes 3, 7, 10, and X. Our composite trait total score was associated with the SNP on BTA10, while the above-mentioned authors found the SNP on BTA5 associated with a total score [28]. In a separate study, [53] found significant SNPs on BTA10, 11, and chromosome X for total score. Our analysis did not cover the X chromosome.

Individual linear traits were associated significantly or near the significance threshold with SNPs located on eleven chromosomes (Table 4). The chromosomes rarely corresponded with other authors. Results that were more or less in common between our study and Cole et al. include a significant SNP on BTA10 for total score and fore udder attachment, as in our analysis. The authors bring detailed information on a large number of SNPs across the whole bovine genome. Wu et al. also found SNPs for the rear leg side view on BTA2, but in our analysis the SNP on BTA2 was only near significance [28,53].

Even when using a less strict method for evaluation, the number of significant SNPs is substantially low compared with other analyses [27,28,53]. Guo et al. reported eleven significant SNPs for angularity, thirteen for body depth, and even twenty-seven for stature [21]. Other authors found five SNPs significantly associated even after the Bonferroni correction with bone quality, six with heel depth, three with rear legs rear view, and six with rear legs side view [54], or thirteen SNPs associated with the udder index [55]. We explain the reasons for low significance in our study as a consequence of the methodology used and, perhaps more importantly, because they were born over a very long period of twenty-five years (1995–2020). Genetic composition changes over time as recombination events change the linkage disequilibrium between markers and causative genes, and different haplotypes occur both within and across populations. The genetic architecture of traits changes considerably across generations, especially under selection and when non-additive effects are present [56]. Genetic changes over time are reflected in the change in variance components, and the decay in accuracy of prediction when the generational distance between the target and training population increases, as observed in chickens and pigs [57,58]. Therefore, old data have a limited impact on the prediction for young animals. Additionally, the phenotype itself has visibly changed over time, reflecting both genetic changes and selection goals. It is therefore possible for a different SNP to become associated with specific traits. However, our analysis exceeded 25,000 animals, which is a considerably larger group than analyzed in other papers [53,54,59]. Hence, we believe that our results could be of interest for the next comparisons.

The genes related to the SNPs or within close proximity are presented in Table 3 and Table 4. Non-coding regions also deserve attention as they may reflect interacting RNAs. GWASs in livestock based on sequencing and high-resolution genotyping have showed that most of the signals associated with composite phenotypic traits are located outside known protein-coding regions in the genome [60]. They analyzed the lncRNAs, while the microRNAs were studied by others, mainly with the aim of describing their role in the control of livestock production traits [61,62] or in the immune response to infectious diseases [63].

Finally, of the five linear composite traits and the twenty linear traits being evaluated, just total score, stature, angularity, and fore udder attachment were found to be significantly associated with SNPs located in BTA10, 20 and 24.

## 4. Conclusions

The objective of this study was to identify the genomic regions involved in the control of body conformation and fitness. Four significant SNPs associated with one linear composite trait and three linear traits were identified. The results of our study can be used to search for causative genes, as well as genomic regions and mutations within the bovine genome that are associated with linear traits to improve the overall body conformation of Czech Holstein cattle.

## Figures and Tables

**Table 1 animals-12-03522-t001:** The number of animals genotyped.

SNP Chip	Genotyped Animals	SNPs
Illumina BovineSNP50 BeadChip V3	15,979	53,218
Illumina BovineSNP50 BeadChip V2	4256	54,609
Euro G MD v1	1408	44,847
Euro G MD v2	1332	51,376
Geenseek GGP 150 k	1313 *	140,668
Geenseek GGP HD_T	1198 *	77,376
Total	25,486	

* only bulls were genotyped.

**Table 2 animals-12-03522-t002:** Descriptive statistics of genomic breeding values (N = 1,772,297) for 25 linear type traits.

Trait	Mean	Variance	SD	Min	Max	h^2^
Angularity	0.49	0.44	0.66	−4.51	4.12	0.29
Body condition score	−0.13	0.09	0.29	−2.11	1.46	0.28
Body depth	0.03	0.09	0.30	−3.36	2.76	0.27
Bone quality	0.25	0.09	0.31	−1.20	1.59	0.26
Central ligament	0.20	0.11	0.32	−2.42	2.96	0.18
Foot angle	0.04	0.06	0.24	−2.91	2.24	0.10
Fore udder attachment	0.23	0.19	0.44	−2.03	3.13	0.24
Front teat placement	0.34	0.25	0.50	−2.24	3.09	0.27
Chest width	−0.01	0.08	0.28	−3.20	2.66	0.18
Locomotion	0.10	0.03	0.18	−1.37	1.81	0.07
Rear legs rear view	0.10	0.04	0.19	−1.99	2.10	0.14
Rear legs side view	0.00	0.06	0.24	−2.68	2.68	0.16
Rear teat placement	0.17	0.08	0.29	−1.47	1.52	0.27
Rear udder height	0.42	0.33	0.57	−3.00	3.33	0.23
Rear udder width	0.28	0.07	0.26	−0.71	1.23	0.19
Rump angle	−0.06	0.16	0.40	−3.01	2.96	0.32
Rump width	−0.06	0.16	0.40	−3.01	2.96	0.40
Stature	0.48	0.46	0.68	−2.72	3.83	0.49
Teat length	−0.05	0.12	0.35	−2.85	3.42	0.32
Udder depth	0.24	0.25	0.50	−2.42	3.23	0.32
Composite dairy character score	0.01	0.00	0.02	−0.09	0.12	0.36
Composite body weight score	0.42	0.23	0.48	−2.04	2.30	0.26
Composite feet and leg score	0.28	0.11	0.33	−2.23	2.70	0.12
Composite udder score	0.38	0.34	0.59	−2.39	2.83	0.20
Composite conformation score	0.56	0.44	0.66	−2.23	3.51	0.25

h^2^—heritability.

**Table 3 animals-12-03522-t003:** The SNP markers with statistically significant (or approaching the significance threshold) associations with linear traits after Bonferroni correction.

Trait	SNP Name	BTA	Position (bp)	rs SNP Name	*p* Value	Gene	Description
**Linear composite traits**							
Dairy capacity composite	ARS-BFGL-NGS-105821	4	58,072,287	rs109967006	1.51×10^−6^	IMMP2L	Mitochondrial membrane peptidase subunit
Feet and legs composite	BTA-52458-no-rs	21	46,984,914	rs41643772	1.46 × 10^−6^	Non-c. seq.	MAP3K12 binding inhibitory protein, 67,979 bp
Total score	ARS-BFGL-NGS-57057	10	101,168,543	rs108945111	9.81 × 10^−7^ *	Non-c. seq.	SPATA7- spermatogenesis associated 7, 17,602 bp
**Linear traits**							
Stature	ARS-BFGL-NGS-112610	24	2,347,580	rs110254857	8.03 × 10^−7^ *	Non-c. seq.	Myelin basic protein—2,224,045 -> 2,328,579, 136,978 bp
Body depth	ARS-BFGL-NGS-83035	6	13,085,315	rs43013615	1.46 × 10^−6^	CAMK2D, intron	Calcium/calmodulin-dependent protein kinase, involvement in regulation of Ca ionts
Angularity	ARS-BFGL-BAC-27938	20	2,997,285	rs208553021	9.96 × 10^−7^ *	RANBP17, intron	RAN-binding protein-17—nucl. transport receptor
Fore udder attachment	Hapmap36641-SCAFFOLD21136_2257	10	47,924,420	rs29020064	4.78 × 10^−8^ **	Non-c. seq.	Nearest gene LOC101902325, 6576 bp

* significant at *p* ≤ 1.42 × 10^−6^; ** significant at *p* ≤ 2.84 × 10^−7^; other SNPs are non-significant but approach the significance threshold; Non-c- seq. non-coding sequence.

**Table 4 animals-12-03522-t004:** The SNP markers with statistically significant (or approaching the significance threshold) associations with linear traits without the Bonferroni correction.

Trait	SNP Name	BTA	Position (bp)	rs SNP Name	*p* Value	Gene	Description
**Linear composite traits**							
Dairy capacity composite	BTB-00025760	1	53,238,076	rs43241717	8.22 × 10^−6^	Non-c. seq.	CD47 (Integrin-ass. prot.), 45830bp; intraflag. transport 5,756,143 bp
	BTA-101359-no-rs	2	56,409,074	rs41570485	9.97 × 10^−6^	LOC101908548, intr.	
	ARS-BFGL-NGS-105821	4	58,072,287	rs109967006	1.51 × 10^−6^	IMMP2L intron	inner mitochondrial membrane peptidase subunit 2
	Hapmap52961-rs29016208	5	69,453,205	rs29016208	8.68 × 10^−6^	Non-c. seq.	Nearest gene chromosome 5 C12orf75 homolog, 200,946 bp
	Hapmap43767-BTA-113302	6	85,646,902	rs41618641	5.64 × 10^−6^	LOC100140029 intr.	
	Hapmap47403-BTA-76048	6	45,153,190	rs41567027	1.00 × 10^−5^	PPARGC1A, intr.	Transcription coactivator regul. gene inv. in energy metabolism
	ARS-BFGL-NGS-103385	7	6,413,319	rs110733477	1.00 × 10^−5^	CHERP, intron	Calcium Homeostasis Endoplasmic Reticulum Protein
	Hapmap43119-BTA-07287	8	104,273,040	rs29026953	9.64 × 10^−6^	WDR31, cod. seq.	Participation in cell processes, cell cycle, apoptosis, signal transduction, gene regulation
	BTB-01915527	18	39,480,387	rs380111366	8.87 × 10^−6^	AP1G1, intron	adaptor rel. protein complex
	Hapmap38047-BTA-101643	22	14,472,226	rs451483233	8.30 × 10^−6^	ABHD5, intron	abhydrolase domain containing 5, lysophosphatidic acid acyltransferase
	ARS-BFGL-NGS-16187	25	7,992,272	rs109583598	6.91 × 10^−6^	Non-c. seq.	C25H16orf72, neg. regulation of signal transduction by p53 class mediator, 35,118 bp
	BTB-00928670	26	19,189,998	rs42092107	7.83 × 10^−6^	R3HCC1L, intr.	
	ARS-BFGL-NGS-106765	28	33,405,497	rs109179573	5.81 × 10^−6^	KCNMA1, intr.	potassium calcium-activated channel subfamily M alpha 1
Feet and legs composite	BTA-37234-no-rs	15	61,893,000		9.70 × 10^−6^		
	BTA-52458-no-rs	21	46,984,914		1.46 × 10^−6^	Non-c. seq.	MAP3K12 binding inhib. prot.
Total score	ARS-BFGL-NGS-57057	10	101,168,543		9.81 × 10^−7^ *	Non-c. seq.	SPATA7- spermatogenesis ass. 7
**Linear traits**							
Stature	ARS-BFGL-NGS-112610	24	2,347,580	rs110254857	8.03 × 10^−7^ *	Non-c. seq.	Myelin basic protein—2,224,045 -> 2,328,579, 136,978 bp
Body depth	ARS-BFGL-NGS-83035	6	13,085,315	rs43013615	1.46 × 10^−6^	CAMK2D, intr.	Calcium/calmodulin-dependent protein kinase
	ARS-BFGL-NGS-4240	23	28,087,630	rs110742604	2.23 × 10^−6^	Non-c. seq.	IER3 (immediate early response 3) and Flotilin1
Angularity	ARS-BFGL-BAC-27938	20	2,997,285		9.96 × 10^−7^ *	RANBP17, intr.	RAN-binding protein-17—nuclear transport receptor
Rump angle	BTB-00087067	2	24,285,533	rs43296861	8.55 × 10^−6^	Non-c. seq.	ITGA6—integrin subulin alpha 6, 67593 bp
Rear legs (rear view)	ARS-BFGL-NGS-37099	9	87,378,346	rs109492604	1.00 × 10^−5^	UST, intron	uronyl-2-sulfotransferase
	BTB-01901596	10	99,234,390	rs43010749	9.23 × 10^−6^	Non-c. seq.	LOC785091, 38,696 bp
Rear legs (side view)	ARS-USMARC-Parent-DQ404152	2	5,306,838	rs29022245	4.58 × 10^−6^	Non-c. seq.	bridging integrator 1, 43,570 bp
	Hapmap53154-ss46527107	19	54,044,562	rs46527107	8.71 × 10^−6^		
Fore udder attachment	Hapmap36641-SCAFFOLD21136_2257	10	47,924,420		4.78 × 10^−8^ *	Non-c. seq.	Nearest gene LOC101902325
Udder depth	ARS-BFGL-NGS-18407	2	77,175,718		3.34 × 10^−6^	CNTNAP5, intr.	Contactin ass. prot. fam. member 5
Rear udder height	BTB-00431144	10	55,323,184	rs43635289	6.47 × 10^−6^	Non-c. seq.	RAB27A RAS oncogene family member, 162,534 bp
Central ligament	Hapmap52274-rs29017133	12	43,799,317	rs29017133	9.39 × 10^−6^	Non-c. seq.	KLHL1, kelch like family member 1, 1,021,225 bp
Rear teat placement	ARS-BFGL-BAC-31288	24	4,273,189	rs42042322	2.25 × 10^−6^	CNDP2	cytosolic non-specific dipeptidase
Rear udder width	Hapmap50787-BTA-80934	8	36,191,988	rs41659555	2.04 × 10^−6^	Non-c. seq.	PTPRD, 32,780 bp, protein PTP signaling molecule regulating cell processes
	ARS-BFGL-NGS-73148	29	40,281,016	rs109455421	3.86 × 10^−6^	Non-c. seq.	LOC100296410, LOC521301, 12,529 bp and 3302 bp

* significant at *p* < 1.00 × 10^−6^; other SNPs are non-significant, but near of the significance threshold; Non-c. seq. non-coding sequence.

## Data Availability

The data used in this study is the property of the Czech-Moravian Breeder´s Corporation (Czech Republic) and the Holstein Cattle Breeder’s Association of the Czech Republic and is therefore not publicly available.

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
