# Peer review of "Genome-Wide Association Study for Body Conformation Traits and Fitness in Czech Holsteins"

_animals, 2022, doi:10.3390/ani12243522_

Round 1
Reviewer 1 Report
1. The significance marking in Table 3 is completely wrong, for instance, the p-values of BTA-101359-no-rs was 9,97E-06 bigger than 1.00E-6 but annotated with *.
2. Please rewrite the result section of Table 3.
3. It is suggested to present the Manhattan and QQ-plot in the paper.
Author Response
Answers to reviewer 1
- The significance marking in Table 3 is completely wrong,for instance, the p-values of BTA-101359-no-rs was 9,97E-06 bigger than 1.00E-6 but annotated with *.
Answer:
The significance marking in the table was corrected. We apologize for error.
- Please rewrite the result section of Table 3.
Answer:
The chapter was rewritten.
- It is suggested to present the Manhattan and QQ-plot in the paper.
Answer:
The Manhattan and QQ-plots were added to the supplementary material (figure S1 and figure S2).
Reviewer 2 Report
Manuscript: animals-1966423-peer-review-v1
A brief summary:
In this manuscript, the authors perform GWAS on body confirmation and fitness traits in Czech Holsteins. The strength of this study is to use more data than previous reports studying dairy cattle. The findings of this study will lead to a better understanding of gene function or a better genomic selection for body confirmation traits in dairy cattle.
General comments:
GWAS studies have been conducted in various dairy breeds for traits similar to those used in this study, but what are the originality and availability of this study? Why does the authors focus on the Czech dairy population? What are the differences from previous studies using Czech dairy breed populations? These should be described in the introduction.
Specific comments:
L148-175: Please present the descriptive statistics of the traits evaluated in this study as Wu et al. have reported (Number of animals, mean, SD, min, max, heritability).
L168: As there is insufficient information to determine the genetic architecture of the population used in this manuscript, authors need to illustrate the extent of linkage disequilibrium in the population used in this study and compare it with previous studies.
L190: The vectors should be shown in bold (same for line 203). Correct the same errors in the entire manuscript. “Y” is not consistent with “y” described in line 191.
L209: Please indicate the first and second steps. It starts from the third in line 217.
L232-235: Does the authors use postGSf90 software and the p-value option to calculate the p-values?
L290: The Figure 1 is not presented in this manuscript.
Table 2 and 3: Why are the descriptions different for overlapping SNP markers detected in GWAS at the two significance levels?
Author Response
Answers to reviewer 2
General comments:
GWAS studies have been conducted in various dairy breeds for traits similar to those used in this study, but what are the originality and availability of this study? Why does the authors focus on the Czech dairy population? What are the differences from previous studies using Czech dairy breed populations? These should be described in the introduction.
Answer:
Rows 148-152: Accepted, the comment was added into the end of the Introduction section:
There are two main dairy breeds in the country, namely Czech Spotted, which is a part of Simmental group, and Holstein. There was not done the GWAS in the Czech dairy cattle, so the paper should fill up the gap.
Specific comments:
L148-175: Please present the descriptive statistics of the traits evaluated in this study as Wu et al. have reported (Number of animals, mean, SD, min, max, heritability).
Answer:
The table with descriptive statistics was created (table 2) and inserted into the text.
L168: As there is insufficient information to determine the genetic architecture of the population used in this manuscript, authors need to illustrate the extent of linkage disequilibrium in the population used in this study and compare it with previous studies.
Answer:
The genetic architecture was described in M&M, chapter Genotyping (rows 190-194). The QQ-plots and genomic inflation factors were added to the supplementary file.
L190: The vectors should be shown in bold (same for line 203). Correct the same errors in the entire manuscript. “Y” is not consistent with “y” described in line 191.
Answer:
The errors in the vectors marking were corrected.
L209: Please indicate the first and second steps. It starts from the third in line 217.
Answer:
Corrected, rows 236, 241
L232-235: Does the authors use postGSf90 software and the p-value option to calculate the p-values?
Answer:
Yes, the p-values were estimated using postGSf90 software version 1.73.
L290: The Figure 1 is not presented in this manuscript.
Answer:
Sorry for our mistake. After this revision, we changed the figure content. You can find QQ plots and Manhattan plots in the supplementary file.
Table 2 and 3: Why are the descriptions different for overlapping SNP markers detected in GWAS at the two significance levels?
Answer:
There was a mistake in the significance marking in Tables 3 and 4. The significance of markers was corrected, and the related text as well.
Reviewer 3 Report
The Authors performed a genome-wide association study (GWAS) on conformation traits using 25,486 genotyped Czech Holsteins, with approximately 35,227 common SNPs for each genotype.
Here are some suggestions that the Authors can consider improving the manuscript.
Line 52: change “vast” with “wide”
Please pay attention to the reference number in bracket in the text for example:
Lines 169-171: please change “……Interbull information from Multiple Across Country Evaluation (MACE) was added according to the methodology from [30].” with “……Interbull information from Multiple Across Country Evaluation (MACE) was added according to the methodology from Přibyl et al [29].” Please verify the reference number on all the manuscript.
Lines 185-186: please change “…. BLUPF90 software suite [31].” with ““…. BLUPF90 software suite [30].” and so on.
Line 191: please change “…where y is the vector of…..” with “……where Y is the vector of…..”.
Line 236: “Result and Discussion” have to be separated in two different Sections as reported in the Instructions for Authors (see “Research Manuscript Sections”).
Tables 2 and 3: please some columns have to be enlarged because are difficult to reading. Please remove the hyperlink in the table.
Author Response
Answers to reviewer 3
Line 52: change “vast” with “wide”
Answer:
Changed.
Please pay attention to the reference number in bracket in the text for example:
Lines 169-171: please change “……Interbull information from Multiple Across Country Evaluation (MACE) was added according to the methodology from [30].” with “……Interbull information from Multiple Across Country Evaluation (MACE) was added according to the methodology from Přibyl et al [29].” Please verify the reference number on all the manuscript.
Lines 185-186: please change “…. BLUPF90 software suite [31].” with ““…. BLUPF90 software suite [30].” and so on.
Answer:
Accepted. There was a mistake in the Reference numbering, namely twice paper No. 1. The error was corrected.
Line 191: please change “…where y is the vector of…..” with “……where Y is the vector of…..”.
Answer:
Corrected.
Line 236: “Result and Discussion” have to be separated in two different Sections as reported in the Instructions for Authors (see “Research Manuscript Sections”).
Answer:
Of course, there are different ways, the separated two sections Results; Discussion, or combination into one. The Instructions for authors allow both. So, the authors suggest leaving the text as it is.
Tables 2 and 3: please some columns have to be enlarged because are difficult to reading. Please remove the hyperlink in the table.
Answer:
The tables were corrected. The legend was moved into the bottom of the table.
Reviewer 4 Report
This is an interesting study, conducted to expected standards. However i have some concerns.
1) The animals were genotyped on different chips, did you do a PCA/MDS to check if there is a bias based on genotyping chip.
2) Did you observe any population structure ? if so how was this accounted ? was the GRM sufficient ?
3) Please provide a manhattan plot and qqplot with genomic inflation factor.
4) There are several studies on body confirmation in holstein, i am curious why your results did not correlate with other studies, considering the flow of genetics across countries. Could it be related to how the traits are measured ?
5) Is it possible to compare the methods of trait measurement between your study and those reported.
6) Can a table be including pointing out which SNPs found significant in your study was previously reported for a similar trait in cattle ?
Author Response
Answers to reviewer 4
1) The animals were genotyped on different chips, did you do a PCA/MDS to check if there is a bias based on genotyping chip.
Answer:
PCA/MDS analysis was not done. Our GWAS was performed based on the same 35,227 SNPs from different chips. We assume that there is no difference between the SNP quality detection among different commercial SNP chips. Animals were genotyped throughout the whole population (bulls, cows) and most of them with the same SNP chip (Illumina BovineSNP50 BeadChip V3). Each chip was used for genotyping of bulls and cows (except high-density Geneseek, where only bulls were genotyped because this chip was used previously before we started cow genotyping). Animals and chips were not especially pre-selected due to this GWAS analysis. These genotypes are routinely used for GEBV prediction, and we include all available chips, but we used only common SNPs for all these chips (we don´t use imputation yet). We add additional information to the M&M about animal and chip selection, rows 190-194.
2) Did you observe any population structure? If so, how was this accounted? was the GRM sufficient?
Answer:
In the GWAS analysis, the whole Holstein population from 1995 to 2020 in the Czech Republic was included. In the relationship matrix, 6 generations were included. The cow's origin is mainly the Czech Republic. Approximately 18 % of bulls are from the domestic population, around 34 % has an origin in USA, 15 % has an origin in Germany. Next, bulls (with a proportion lower than 15 %) are from Canada, Netherland, France, Italy, Spain and Slovak republic. Relatedness between Holstein subpopulations from different countries is high but the bull´s subpopulation shows the highest genetic similarity with the subpopulation of bulls from the USA (probably due to import). The effective population size of Holstein in Czech Republic is Ne = 202 (only for cows Ne = 186; for bulls Ne = 129). In this Holstein population, the connectedness between animals is good (there are no genetically separated animals). Furthermore, any such structure would appear both by pedigree (expected relatedness) in relationship matrix A and genomic relationship matrix G (true relatedness). All bulls used in the mating must be genotyped, so we covered the whole male population used in this population.
The only inconvenience could occur in the linear type trait evaluation during the years 1995-2020 because the population was changing through time but we included all these phenotypes in the study. This fact could cause the low association and significance of some SNPs with linear type traits. Nevertheless, the most of included genotypes (aprox. 98%) are from young animals.
3) Please provide a Manhattan plot and qq-plot with genomic inflation factor.
Answer:
Manhattan plots and qq-plots were added in the supplementary file.
4) There are several studies on body confirmation in Holstein, I am curious why your results did not correlate with other studies, considering the flow of genetics across countries. Could it be related to how the traits are measured?
Answer:
We tried to discuss the reasons in the Results and Discussion section, the 3rd paragraph from the bottom (We explain the reasons…….associated with specific traits.).
Our results did not correlate to other studies, but our study was done on a large population; 1.7 million animals with approximately 700 thousand of phenotypes during years 1995-2020 and 25,486 genotyped Holstein individuals (5,419 siress and 20,067 cows) were included. We assume this population is more complex than the population in other studies. For example, Wu et al. 2013 included only 1,314 Holstein cows, Bouwman et al. 2018 included 1,565 Holstein individuals, Zhang et al. 2017 included 3,325 Holstein individuals, Lu et al. 2021 included 1,730 Holstein cows and Nazar included 1,000 holstein cows from 4 farms. So, we assume that our results are affected by the inclusion of the large part of the whole Holstein population in our country which is also affected by the sires from foreign countries.
Looking at the GWAS studies reported by Wu et al. 2013, the results did not correlate to studies reported by Nazar et al. 2022 (doi: 10.3390/ani12192542), Abdalla et al. 2021 and Lu et al. 2021 too, even they are all performed on the Chinese Holstein cattle population. For the same linear type traits (fore udder attachment, central ligament, rear attach height and width, loin strength, rump angle, bone quality and rear leg side view), they observed different significant SNPs which are even on the different chromosomes (except rump angle on chromosome 7 reported by Wu et al. 2013 and Lu et al. 2021 (doi: 10.3390/ani11071927). Based on these results, we do not find the difference between results critical.
5) Is it possible to compare the methods of trait measurement between your study and those reported.
Answer:
Linear type trait evaluation is unified for many countries which are under the methodology of World Holstein Friesian Federation (1 to 9 scale). The study reported by Wu et al. (2013) and by Lu et al. (2021) and Nazar et al. (2022) also used the score from 1 to 9. We suppose that the methodology is the same or very close to our methodology.
6) Can a table be including pointing out which SNPs found significant in your study was previously reported for a similar trait in cattle?
Answer:
Yes, such extension would be possible. But it would complicate the tables and worsen the clear arrangement. So, the authors suggest not to add the citations.
Reviewer 5 Report
Dear authors,
This study analyzes a total of 25,000 Czech Holsteins with GWAS to relate the SNPs found with different traits, including dairy capacity composite, feet and legs composite, stature, body depth, and angularity, among others. The results obtained showed significant values for 31 SNPs. These results are very interesting to breeding improvement of cows. This paper is well written and structured, the introduction provides sufficient background and includes relevant references, the cited references are relevant to the research, the research design is appropriate, the methods are adequately described, the results are clearly presented, and the conclusions are supported by the results. Only minor comments:
- Line 136: Authors should type citations in order.
- Line 290: the figure 1 is missing.
- Line 298: add the name of the authors of reference 49.
- Lines 303-305: add reference and data of Cole et al., at the last of the sentence.
1.- The main question addressed by the research is to evaluate the results obtained with genome-wide association study (GWAS) on conformation traits using 25,000 cows.
2.- The topic is relatively original, because more studies related to GWAS and cows has been realized.
3.- The manuscript adds to the subject area news SNPs related to the conformation traits in cows.
4.- The authors could improve the analysis related to other productive traits, more relevant to animal production.
5.- The conclusions are consistent with the evidence and arguments presented.
6.- The references are appropriate.
Author Response
Answers to reviewer 5
Line 136: Authors should type citations in order.
Answer:
There was an error in the numbering, namely twice reference No. 1. The error was corrected.
Line 290: the figure 1 is missing.
Answer:
Sorry for our mistake. After this revision, we changed the figure content. You can find QQ plots and Manhattan plots in the supplementary file.
Line 298: add the name of the authors of reference 49.
Answer:
Revised.
Lines 303-305: add reference and data of Cole et al., at the last of the sentence.
Answer:
The text was revised.
Round 2
Reviewer 2 Report
Although the authors responded well to the reviewers’ comments, a few minor changes are still required. The method of estimating heritability for traits should be described. In addition, please show the standard errors of heritability estimates in Table 2.
Author Response
Answers to reviewer 2
Reviewer comment: Although the authors responded well to the reviewers’ comments, a few minor changes are still required. The method of estimating heritability for traits should be described. In addition, please show the standard errors of heritability estimates in Table 2.
Authors comment:
Dear reviewer, the genetic parameters and heritability coefficients were not calculated in this study. We were based on the results of the study Němcová et al.2011 and Zavadilová et al. 2014. In the study of Němcová et al. 2011, the genetic parameters and heritabilities were estimated. The approximate standard errors of h2 ranged from 0,0050 to 0,0122. In the study of Zavadilová, et al. 2014, the model equation for genomic breeding prediction was developed. Our research followed these studies. We add this information to the text – lines 205 to 208:
,,Our study followed the studies of Němcová et al. [32] where the genetic parameters and heritability coefficients were estimated and Zavadilová et al. [33] where the single-step genomic evaluation for linear type traits was developed.”
And also we add these references:
- Němcová, E.; Štípková, M.; Zavadilová, L. Genetic parameters for linear type traits in Czech Holstein cattle. Czech. J. Anim. Sci. 2011, 56 (4), 157-162.
- Zavadilová, L.; Přibyl, J.; Vostrý, L.; Bauer, J. Single-step genomic evaluation for linear type traits of Holstein cows in Czech Republic. Anim. Sci. Pap. Rep. 2014, 32 (3). 201-208.

Reviewer 4 Report
Thank you for responding to my comments and making the changes suggested.
Author Response
.